# Phenotypic and Functional Characterization of NK Cells in αβT-Cell and B-Cell Depleted Haplo-HSCT to Cure Pediatric Patients with Acute Leukemia

**DOI:** 10.3390/cancers12082187

**Published:** 2020-08-05

**Authors:** Raffaella Meazza, Michela Falco, Fabrizio Loiacono, Paolo Canevali, Mariella Della Chiesa, Alice Bertaina, Daria Pagliara, Pietro Merli, Valentina Indio, Federica Galaverna, Mattia Algeri, Francesca Moretta, Natalia Colomar-Carando, Letizia Muccio, Simona Sivori, Andrea Pession, Maria Cristina Mingari, Lorenzo Moretta, Alessandro Moretta, Franco Locatelli, Daniela Pende

**Affiliations:** 1Laboratory of Immunology, Istituto di Ricovero e Cura a Carattere Scientifico (IRCCS) Ospedale Policlinico San Martino, 16132 Genoa, Italy; raffaella.meazza@hsanmartino.it (R.M.); fabrizio.loiacono@hsanmartino.it (F.L.); pcanevali@gmail.com (P.C.); nataliacolomar@gmail.com (N.C.-C.); mariacristina.mingari@unige.it (M.C.M.); 2Laboratory of Clinical and Experimental Immunology, Integrated Department of Services and Laboratories, IRCCS Istituto Giannina Gaslini, 16147 Genoa, Italy; michelaemma.falco@gmail.com; 3Department of Experimental Medicine, Centre of Excellence for Biomedical Research, University of Genoa, 16132 Genoa, Italy; Mariella.DellaChiesa@unige.it (M.D.C.); simona.sivori@unige.it (S.S.); alemoret@unige.it (A.M.); 4Department of Hematology/Oncology, IRCCS Ospedale Pediatrico Bambino Gesù, 00146 Rome, Italy; aliceb1@stanford.edu (A.B.); daria.pagliara@opbg.net (D.P.); pietro.merli@opbg.net (P.M.); federica.galaverna@opbg.net (F.G.); mattia.algeri@opbg.net (M.A.); francina.moretta@gmail.com (F.M.); franco.locatelli@opbg.net (F.L.); 5Giorgio Prodi Interdepartmental Center for Cancer Research—CIRC, University of Bologna, 40138 Bologna, Italy; valentina.indio2@unibo.it (V.I.); andrea.pession@unibo.it (A.P.); 6Department of Experimental Medicine, University of Genoa, 16132 Genoa, Italy; letiziamuccio@hotmail.it; 7Department of Pediatrics, University of Bologna, 40138 Bologna, Italy; 8Department of Immunology, IRCCS Ospedale Pediatrico Bambino Gesù, 00146 Rome, Italy; lorenzo.moretta@opbg.net; 9Department of Gynecology/Obstetrics and Pediatrics, Sapienza University, 00185 Rome, Italy

**Keywords:** NK cells, haploidentical hematopoietic stem cell transplantation (HSCT), killer Ig-like receptors (KIR), NK alloreactivity, pediatric acute leukemia, human cytomegalovirus (HCMV)

## Abstract

NK cells can exert remarkable graft-versus-leukemia (GvL) effect in HLA-haploidentical hematopoietic stem cell transplantation (haplo-HSCT). Here, we dissected the NK-cell repertoire of 80 pediatric acute leukemia patients previously reported to have an excellent clinical outcome after αβT/B-depleted haplo-HSCT. This graft manipulation strategy allows the co-infusion of mature immune cells, mainly NK and γδT cells, and hematopoietic stem cells (HSCs). To promote NK-cell based antileukemia activity, 36/80 patients were transplanted with an NK alloreactive donor, defined according to the KIR/KIR-Ligand mismatch in the graft-versus-host direction. The analysis of the reconstituted NK-cell repertoire in these patients showed relatively high proportions of mature and functional KIR^+^NKG2A^−^CD57^+^ NK cells, including the alloreactive NK cell subset, one month after HSCT. Thus, the NK cells adoptively transfused with the graft persist as a mature source of effector cells while new NK cells differentiate from the donor HSCs. Notably, the alloreactive NK cell subset was endowed with the highest anti-leukemia activity and its size in the reconstituted repertoire could be influenced by human cytomegalovirus (HCMV) reactivation. While the phenotypic pattern of donor NK cells did not impact on post-transplant HCMV reactivation, in the recipients, HCMV infection/reactivation fostered a more differentiated NK-cell phenotype. In this cohort, no significant correlation between differentiated NK cells and relapse-free survival was observed.

## 1. Introduction

Hematopoietic stem cell transplantation from an HLA-haploidentical donor (haplo-HSCT), represented by a relative sharing half of HLA alleles (i.e., one haplotype) with the recipient, is a suitable option for patients with hematologic diseases in urgent need of an allograft, but lacking an HLA-matched donor [1]. Haplo-HSCT became feasible in the 1990s by intensifying the conditioning regimen to prevent graft rejection, and by the infusion of “megadoses” of CD34^+^ purified cells, thus depleting T cells responsible of the occurrence of graft-versus-host disease (GvHD) [2]. However, this graft manipulation causes a prolonged lymphopenia and a delayed immune reconstitution, leading to severe opportunistic infections. The graft-versus-leukemia (GvL) effect of CD34^+^ haplo-HSCT (T-cell depleted) mainly relies on NK cells, which are the first lymphocyte subset reconstituting into the patient after transplantation. A better clinical outcome, in terms of leukemia-free survival (LFS), has been associated with donor NK alloreactivity versus the recipient both in adults and in pediatric patients [3,4,5]. Conversely, controversial results have been described in un-manipulated (T-cell replete) haplo-HSCT, with studies showing that NK alloreactivity was associated with a better [6,7,8] or worse [9,10,11] clinical outcome. The conflicting results might depend on the different transplantation protocols and the models to define NK alloreactivity [12]. We believe that the most reliable way to evaluate NK alloreactivity should take into consideration the HLA incompatibility between donor and recipient, the donor killer Ig-like receptor (KIR) repertoire, and KIR education [13]. The inhibitory KIRs (iKIRs) are specific for allotypic determinants shared by distinct groups of HLA class I molecules (referred to as KIR-ligands, KIR-L) [14,15,16]. They include KIR2DL1, specific for HLA-C^Lys80^ allotypes carrying C2 epitope (i.e., HLA-C2), KIR2DL2/L3 recognizing HLA-C^Asn80^ allotypes carrying C1 epitope (i.e., HLA-C1), but also with low-affinity HLA-C2, and KIR3DL1 specific for HLA-B and HLA-A molecules sharing the Bw4 public epitope (i.e., HLA-Bw4). Thus, donor NK alloreactivity can be identified according to the presence of donor iKIR(s) specific for KIR-L(s) present in the donor and absent in the recipient, following the rules of the “missing self” recognition (i.e., KIR/KIR-L mismatch in the GvH direction). This takes into account the education process during NK-cell development, through which NK cells acquire functional competence only when they express at least one inhibitory NK receptor (iNKR) specific for the self HLA class I [17,18]. To ensure self-tolerance, an opposite effect of activating KIR (aKIR), when engaged by their cognate ligands, occurs. In particular, the expression of KIR2DS1, specific for HLA-C2, causes NK-cell hypo-responsiveness in HLA-C2 homozygous donors, and KIR2DS1^+^ NK cells are educated only in HLA-C1/Cx individuals [19]. In addition to KIR, other HLA-specific NKR are the inhibitory CD94:NKG2A and the activating CD94:NKG2C heterodimers, both recognizing HLA-E molecules [20]. NK-cell functional activity is triggered by activating receptors and co-receptors that interact with specific ligands on tumor target cells. Relevant receptors involved in anti-leukemia activity are NKp46, NKp30, and NKp44 (collectively called natural cytotoxicity receptors, NCR), whose surface-bound ligands are still not completely defined, NKG2D interacting with MICA/B and ULBPs, and DNAM-1 recognizing PVR (CD155) and Nectin-2 (CD112) [21,22,23,24].

Among different individuals, extremely variegated NK-cell phenotypic repertoires can be observed, primarily depending on *KIR* and *HLA* gene content, but also on the stochastic expression of NKR and their clonal distribution [25,26]. Moreover, additional factors have a relevant influence. Among viral infections, HCMV greatly influences the NK-cell phenotype, promoting the expansion of highly differentiated NK cells with “adaptive” features, characterized by the expression of NKG2C, self-iKIR (primarily KIR2DL), the marker of terminally differentiated stage CD57, and the lack of NKG2A [27,28,29]. The role of HCMV in accelerating NK-cell maturation has been described in patients after allogeneic HSCT, in different transplantation settings [30,31,32].

As documented in CD34^+^ haplo-HSCT recipients, the NK cells derived early from HSC display an immature phenotype, characterized by CD56^bright^KIR^−^NKG2A^+^ expression; the emergence of fully functional KIR^+^ mature NK cells, including the alloreactive NK cells, may require at least eight weeks, this resulting into a delay of the NK cell-associated GvL effect [13,33]. Notably, the use of a novel graft manipulation method, based on the selective depletion of αβT cells and B cells, allows the infusion, in addition to HSC, of immunocompetent cells such as mature, donor-derived NK, γδT, and myeloid cells. In addition, in this setting of selective T-cell depletion, no post-transplant pharmacological immune suppression is given, and NK cells can promptly exert an immediate anti-leukemia effect after transplantation, before the wave of NK cells differentiating from donor hematopoietic precursors emerges. Here, we analyzed the cohort of patients transplanted from αβT/B-cell depleted haplo-HSCT (NCT01810120), whose clinical outcome has been already described [34], providing new insights on NK-cell receptor repertoire of donors and transplanted patients.

## 2. Results

### 2.1. Criteria for Donor Selection

To select the most suitable donor, when alternative donors (e.g., both parents) were available (58 out of 80 cases), we considered several features found to be correlated, by either in vitro assays and/or clinical studies, with a better anti-leukemia potential. These criteria, defined by genetic and phenotypic analyses, included: (a) presence of NK alloreactivity (i.e., KIR/KIR-L mismatch in GvH direction) and larger size (i.e., ≥5%) of the alloreactive subset [3,33], (b) presence of a *KIR* B/x genotype especially with B content value ≥2 [35], (c) presence of KIR2DS1 [36,37], (d) higher absolute number of NK and γδT cells [38], and (e) higher expression of NKp46 [21] and presence of NKG2C [32]. Table 1 and Appendix A report patient and donor characteristics including: type of disease (acute lymphoblastic leukemia, ALL, or acute myeloid leukemia, AML), presence, and type of donor NK alloreactivity, *KIR* genotype, B content value, and patient clinical outcome.

All donors were analyzed combining genetic and phenotypic approaches. An example, comparing the parents of a patient, is shown in Appendix A. Based on *KIR*-ligand and *KIR* genotype analyses, only the mother could be considered a NK alloreactive donor. Flow-cytometry analysis using different anti-KIR mAb combinations allowed the detection of surface expressed iKIR and aKIR, and the evaluation of the alloreactive subset. In addition, the expression of NKp46, as well as of NKG2C in relation to NKG2A, KIRs, and CD57, was also tested. In this case, the mother was preferred as donor for the transplant, mainly because of the presence of NK alloreactivity and higher B content value.

In the next two sub-sections, the relevance of NK alloreactivity and activating NK cell receptors will be analyzed in detail.

#### 2.1.1. Advantage of NK Alloreactivity in the Anti-Leukemia Effect

The presence of NK alloreactivity represented the first criterion considered for donor selection, since alloreactive NK cells, expressing as inhibitory receptor(s) only the educated iKIR(s) which do not recognize any ligand on recipient cells, can display a higher anti-leukemia activity and thus potentially exert GvL effect [21,33]. Three types of NK alloreactivity were identified based on donor self-iKIR and mismatched KIR-L. They included Allo C1 (mediated by KIR2DL2/L3 and mismatched HLA-C1), Allo C2 (KIR2DL1 and mismatched HLA-C2), and Allo Bw4 (KIR3DL1 and mismatched HLA-Bw4). In our cohort, 45% of the patients were transplanted from an NK alloreactive donor (Table 1 and Appendix A). To functionally assess the advantage of NK alloreactivity, we performed rapid degranulation assays. We focused on Allo C2, since KIR2DL1 displays a more stringent recognition of HLA-C allotypes, and a greater number of HLA-C1/C1 than HLA-C2/C2 leukemia blasts, in addition to the pediatric leukemia cell line NALM-16 (HLA-C1 hemizygous) were available (Appendix A). NK cells derived from HLA-C1/C2 donors were significantly more efficient than NK cells from HLA-C1/C1 donors against HLA-C1/C1 pediatric ALL (Figure 1A, left). Importantly, considering the activity of different subsets in the case of HLA-C1/C2 effector cells, the alloreactive NK cell subset (i.e., cells expressing KIR2DL1 as the only inhibitory receptor, single positive KIR2DL1, sp2DL1) displayed a significantly higher degranulation capability than other subpopulations (Figure 1A, right). As shown in a representative experiment (Figure 1B), sp2DL1 was the most efficient subset against HLA-C1/C1 pediatric ALL-1 and AML-2, both characterized by high surface expression of HLA-C as detected by DT9 mAb (Appendix A). NALM-16, characterized by low expression levels of HLA class I molecules, appeared to trigger NK-cell degranulation, but still the advantage of the alloreactive subset could be detected (Figure 1B and Appendix A).

#### 2.1.2. Relevance of Activating NK Cell Receptors

In the absence of NK alloreactivity, donor selection criteria were primarily based on the higher B content value, described to correlate with a better LFS [35,39]. Among aKIRs, the presence of KIR2DS1 was positively considered, particularly in HLA-C1^+^ donor and HLA-C2^+^ patient pairs, implying its education and its capability to recognize the ligand on leukemia blasts (termed “educated and useful”, E/U) [19,33]. Thus, 45% or 35% of transplants were performed from donors characterized by *KIR* B content value ≥ 2 or by the presence of KIR2DS1 E/U, respectively (Appendix A). In addition, assuming a relevant role of NKG2C receptor during HCMV reactivation [30,32], whenever possible, we avoided choosing donors characterized by *NKG2C^del/del^* genotype. In this cohort, only 3 out of 80 donors were *NKG2C^del/del^*.

Since within the activating receptors involved in natural cytotoxicity NCRs play a major role in leukemia recognition [21], we examined the NK-cell expression of NKp46, which may vary among different donors, privileging those with higher surface density. We analyzed various pediatric primary leukemia cells as targets to define the involvement of different activating pathways in NK-mediated killing. Effector cells were represented by polyclonal activated NK cells, selected according to their killing capacity. As shown in Figure 1C, mAb-mediated blocking of NCR, as compared to NKG2D and DNAM-1, induced a higher inhibition of lysis of most primary leukemia blasts, both BCP-ALL, T-ALL, and AML. This confirmed the primary role of NCR/NCR-L interaction in NK-cell-mediated anti-leukemia activity. The ULBP1^+^ ALL-2, in which NKG2D-mediated killing was primarily effective, represented an exception. The NALM-16 leukemia cell line was highly susceptible to NK-cell lysis by the engagement of NKG2D, NCR, and DNAM-1 triggering receptors, in order of relevance, consistent with their expression of all NKG2D-L and DNAM-1-L (Appendix A and Appendix A).

### 2.2. Infused NK Cells Persist and Are Functional up to One-Month Post-Transplant

In order to assess the possible contribution of the NK cells adoptively infused with the graft, we studied the pattern of receptor expression on NK cells at an early phase (approximately 1 month) after αβT/B-depleted haplo-HSCT. In most instances, CD94:NKG2A, the inhibitory receptor characterizing NK cells at the immature stages, was expressed on a fraction of NK cells. We stratified the patients in two groups according to the number of infused NK cells, namely higher or lower than the median value (i.e., 34.6 × 10^6^/kg body weight). Patients infused with high NK cell numbers displayed a significantly lower % NKG2A^+^ NK cells, as compared to those infused with low NK cell numbers (*p* < 0.05) (Figure 2A). For comparison, we also analyzed the reconstitution in a cohort of patients after CD34^+^ haplo-HSCT, which is typically characterized by NK cells with an immature phenotype one-month post-transplant [33]. Notably, only the αβT/B-depleted haplo-HSCT patients receiving a high NK cell dose greatly differed from CD34^+^ haplo-HSCT in terms of NKG2A expression (*p* < 0.001). Thus, the infusion of high NK cell doses resulted in higher frequency of mature NK cells defined as KIR^+^NKG2A^−^ (Figure 2B). If the graft contained high NK cell numbers with adaptive phenotype (i.e., high percentages of NKG2C and CD57), the early-reconstituted NK-cell repertoire of the patient showed a pattern of receptor expression very similar to the respective donor. A representative case is shown in Figure 2C. Remarkably, this reflected also in the identification of the alloreactive NK cell subset in the early phase after transplantation (Table 1 and Figure 2D). Importantly, the anti-leukemia activity of NK cells was also preserved. Indeed, NK cells from αβT/B-cell depleted haplo-HSCT patients at one month after the transplant displayed similar degranulation capacity against K562 as compared to NK cells from healthy donors (Figure 2E). In addition, the sp2DL1 NK cell subset of patients transplanted from Allo C2 donors was the most efficient against NALM-16, similarly to NK cells from healthy donors (Figure 2F and Appendix A).

Altogether, these data highlight that engrafted mature and functional NK cells persist in the peripheral blood of the recipient for at least one month.

### 2.3. Analysis of NK Alloreactivity in the Reconstituted Repertoire after Transplantation

If the detection of mature NK cells at the 1st month post-transplant mainly relies on the phenotype and dose of engrafted NK cells, at the following time points, the NK-cell repertoire can be complemented and thus influenced by newly differentiated NK cells from immature precursors. We compared the presence of alloreactive NK cell subset in the reconstituted NK cell repertoire of the recipients with the respective donors, and analyzed the functional activity of NK cells after transplantation.

#### 2.3.1. Dynamics of the Alloreactive NK Cell Subset

Table 1 summarizes the 36 cases transplanted from NK alloreactive donors, grouped by the type of alloreactivities: 15 cases of Allo C1 (including two cases with additional Bw4 mismatch), 14 cases of Allo C2, and seven cases of Allo Bw4. Among the activating KIRs, the presence of KIR2DS1 and KIR2DS2 has been reported. In particular, KIR2DS1 E/U was favored in Allo C1 and Allo Bw4 groups (present in 10 out of 22 cases). The presence of KIR2DS2 leads to an imprecise definition of the size of alloreactive subset for Allo C2 and, possibly, Allo Bw4, since it can be co-expressed with KIR2DL1 and KIR3DL1, respectively, but these cells cannot be quoted as alloreactive (due to the lack of an antibody recognizing KIR2DL2/L3 and not KIR2DS2). We analyzed the frequency of alloreactive NK cells present in donors and patients at post-transplant time ranging from 3 to 6 months (3–6 M), in comparison to one month (1 M) (Table 1). Figure 3A shows the comparison of the percentage of the alloreactive NK cell subsets detected in the donors and in the 3–6 M recipients.

#### 2.3.2. Influence of HCMV Reactivation on Alloreactive NK-Cell Phenotype

The post-transplant HCMV reactivation, which is known to promote the differentiation of mature KIR^+^NKG2A^−^ [31,40], possibly containing alloreactive cells, was also evaluated (Figure 3A). In the Allo C1 group, the alloreactive subset (i.e., sp2DL2/3) was substantially preserved post-transplantation and it appeared to be sustained by HCMV reactivation. A strong reduction of alloreactive subset was observed only in two patients (UPN35 and UPN40), who received a graft containing low numbers of NK cells and did not experience HCMV reactivation. In the Allo C2 group, the size of the alloreactive subset almost always appeared reduced in the reconstituted patient repertoire in comparison to the respective donors, and only one exception (UPN24) was observed. Nevertheless, in 6 out of 13 cases analyzed, we observed percentages ≥ 5 (value arbitrarily chosen as threshold for a substantial size of the alloreactive subset). In addition, the possible under-estimation due to the presence of KIR2DS2 should be considered. Notably, HCMV reactivation, observed in three patients, was associated with a reduction of the sp2DL1 subpopulation, possibly due to the lack of NKG2C^+^ cell subset increase (i.e., UPN23 and UPN43) or the expansion of NKG2C^+^KIR2DL2/L3^+^ NK cells, limiting alloreactivity (i.e., UPN71). In the Allo Bw4 group, the presence of a substantial post-transplant alloreactive subset was only observed in UPN51, who did not experience HCMV reactivation. In UPN12 and UPN26 (HLA-C1^+^ donors and recipients), an expansion of NKG2C^+^KIR2DL2/L3^+^ NK cells, possibly induced by HCMV reactivation, resulted in sp3DL1 cell subset reduction.

#### 2.3.3. Anti-Leukemia Activity

We tested the functional activity of NK cells derived from representative transplanted patients against primary leukemia blasts in degranulation assays. In UPN7 and UPN29, representative of Allo C1 group, NK cells degranulated when co-cultured with ALL-2 (characterized by HLA-C2/C2), and the alloreactive subset was the most efficient (Figure 3B). In the Allo C2 group, a good degranulation capacity, especially from the alloreactive subset, was observed in UPN58 (and to a less extent in UPN43) NK cells against the HLA-C1/C1 leukemias, AML-2 and ALL-1 (Figure 3C).

### 2.4. Evaluation of Differentiated vs. Naïve NK-Cell Repertoires and Possible Correlation with the Clinical Outcome

Similar to the approach described by Bjorklund et al. [41], we performed an unsupervised hierarchical clustering of relevant NK-cell immunophenotyping data of both donors (Appendix A) and recipients at 3–6 months after transplant (Figure 4). The frequencies of CD56^bright^, and, among the CD56^dim^, of NKG2A^+^, CD57^+^, NKG2C^+^, and KIR^+^NKG2A^−^ cells have been evaluated. In these analyses, cases with *NKG2C*^del/del^ donors (*n* = 3) have been excluded. Two main NK cell clusters, associated with a naïve (high NKG2A and/or CD56^bright^) or more differentiated (high NKG2C, KIR^+^NKG2A^−^ and/or CD57^+^) pattern, were identified in both donors and recipients.

We could not prove any association between the different donor clusters and HCMV reactivation after haplo-HSCT, suggesting that the differentiated phenotype of donor NK cells represents neither a predisposing nor a protecting feature (*p* = 0.79) (Figure 5A). In contrast, the evaluation of the recipients’ repertoire led to the evidence that the cluster with differentiated phenotype was associated with HCMV reactivation (*p* = 0.002) (Figure 5C). These data are consistent with the notion that HCMV reactivation accelerates NK-cell differentiation, thus proving the accuracy of this clustering approach. Therefore, we assessed whether a certain cluster identified in both donors and transplanted patients might be associated with the clinical outcome. However, we could not find any significant correlation with the overall survival (OS), LFS, and cumulative incidence of relapse (Figure 5B,D). Stratifying by disease type of the transplanted patients (ALL and AML), we could observe an opposite trend toward a better clinical outcome, considering the donor cluster: differentiated for ALL, and naïve for AML patients (Appendix A).

## 3. Discussion

Several platforms of allogeneic HSCT can be employed to treat leukemia patients, including the use of different type of donor with diverse HLA matching (HLA-identical sibling, SIB; HLA-matched unrelated volunteer, URD; umbilical cord blood, UCB; or haploidentical), different stem cell source (umbilical cord blood unit, UCB; bone marrow, BM; or mobilized peripheral blood, PBSC), various preparative conditioning regimen (reduced intensity, RIC; or myeloablative, MAC), graft manipulation strategies and pharmacological GvHD prophylaxis [1,42]. All these variables, along with the patient age (pediatric or adult), may influence the post-HSCT NK-cell reconstitution affecting both phenotypic and functional features. For children with high-risk leukemia in urgent need of a transplant, both parents can represent readily available haploidentical donors. However, to circumvent the risk of graft rejection and GvHD related to the high degree of HLA mismatch intrinsic to the haploidentical setting, different strategies have been exploited, including the use of post-transplant immune suppression or ex vivo graft manipulation to obtain an extensive T-cell depletion [13,43,44,45]. The haplo-HSCT platform offered to the cohort of pediatric patients analyzed in this study took advantage of an innovative strategy based on the selective depletion of αβT lymphocytes (major responsible of GvHD) and CD19^+^ B cells (responsible of post-transplant EBV-related lymphoproliferative disorders) [46]. Notably, this procedure allows for overcoming the delayed immune reconstitution, which is typical of CD34^+^ haplo-HSCT [47]. In fact, the transfer to the recipient of high numbers of HSCs, CD34^−^ intermediate precursors and of myeloid cells, including monocyte/dendritic cells, can contribute to the low risk of non-relapse mortality (NRM) [34]. Additionally, the presence of mature NK and γδT cells can protect against life-threatening infections and leukemia relapse [13]. Lastly, the lack of any post-transplant GvHD prophylaxis allows the persistence in the circulation of the infused cells. The study of kinetics and function of reconstituting γδT cells in pediatric patients enrolled in the same clinical trial has been already described [48]. Briefly, γδT cells represent the predominant T-cell population during the first weeks after transplantation, derived from cells infused with the graft and expanded in vivo, particularly the Vδ1 cells in patients reactivating HCMV. The γδT cells were shown to exert an anti-leukemia activity, which could be potentiated especially in Vδ2 cells upon ex-vivo exposure to zoledronic acid. Our present study represents the first in-depth characterization of the post-transplant NK-cell dynamics, through the analysis of phenotypic receptor repertoire, alloreactive NK cell subset, and anti-leukemia activity. One month after αβT/B-cell depleted HSCT, especially when high numbers of NK cells had been infused, we could detect a relatively high proportion of CD56^dim^KIR^+^NKG2A^–^ NK cells, and sizeable alloreactive and/or adaptive subsets, resembling the donor NK-cell phenotypic repertoire (Figure 2). Remarkably, we also demonstrated their anti-leukemia activity. These findings highly suggest that donor derived NK cells, adoptively infused with the graft, persist in recipient peripheral blood. This represents a remarkable peculiarity of the αβT/B-cell depleted haplo-HSCT. In fact, early after either CD34^+^ haplo-HSCT [33] and T cell-replete haplo-HSCT [49] (which is based on the infusion of an unmanipulated graft and post-transplant high-dose cyclophosphamide (PT-Cy) followed by other immunosuppressive drugs) NK cells exclusively display an immature phenotype. In the PT-Cy platform, it has been recently documented that infused NK cells first proliferate in response to the high systemic levels of IL-15 and then become sensitive to Cy-mediated killing, resulting in early elimination of all mature NK cells, including the alloreactive subset [49]. Similarly, we observed a dominant CD56^bright^KIR^−^NKG2A^+^ NK cell phenotype in both adult [50] and pediatric (unpublished) patients given PT-Cy haplo-HSCT.

In αβT/B-depleted haplo-HSCT, we speculated that the choice of a donor whose NK-cell repertoire contains alloreactive NK cells represent a relevant advantage to boost anti-leukemia activity. Indeed, in the absence of an “off” signal, these NK cells can lyse target cells expressing ligands of triggering receptors, mainly NCR-L, NKG2D-L, and DNAM-1-L [21,23,33]. Here, we compared the functional capability of different NK cell subsets on the basis of rapid degranulation assays against leukemia blasts as target cells. The use of primary leukemia blasts represents a major technical challenge. Nevertheless, especially in the combination of NK cells from HLA-C1/C2 donors and leukemia from HLA-C1/C1 patients, we could clearly show that sp2DL1 NK cells (i.e., Allo C2 subset) are endowed with the highest effector function. This was observed by analyzing NK cells from both donors and transplanted patients. These findings are consistent with previous functional data obtained with NK cell clones in chromium-release assays [33]. Leukemia cell lines such as K562 and NALM-16, characterized by absent/low HLA class I expression and a wide panel of ligands for triggering receptors, can stimulate NK cell cytotoxicity and degranulation with higher efficiency than that induced by primary leukemia blasts. By comparing the HLA-C1 hemizygous NALM-16 with HLA-C1/C1 primary leukemia blasts as target cells, a marked difference could be detected. Remarkably, sp2DL1 cells exceeded the other NK cells in degranulation capacity even with NALM-16 target cells. Thus, the use of NALM-16 allowed for documenting the superior functional activity of sp2DL1 NK cells isolated from patients at early time points after HSCT from Allo C2 donor. It is conceivable that a similar advantage may be detected also with primary leukemia blasts. While the surface detection of the NKG2D-L MICA and ULBPs, of the DNAM-1-L PVR and Nectin-2, and of the NKp30-L B7-H6 is possible thanks to the use of specific mAb, reagents are missing to detect other NCR-L. In this context, receptor blocking experiments suggested a primary role of NCR (in particular NKp46) in NK-cell recognition and killing of different pediatric primary leukemia. On the basis of these in vitro data, NK cell populations characterized by a large size of the alloreactive subset and by a NCR^bright^ phenotype are likely to exert an optimal anti-leukemia activity. In addition, B/x *KIR* genotype and high B content value, the presence of KIR2DS1 E/U, and the presence of NKG2C, indicating a receptor repertoire well equipped with activating receptors, were positively considered in our donor selection criteria [4,12,35,37,39,51,52]. Unexpectedly, no significant correlation with LFS was observed considering donor NK alloreactivity, *KIR* genotype, B-content score and KIR2DS1 E/U [34]. As a general comment, since αβT/B-depleted haplo-HSCT is an extremely successful procedure, with LFS probability around 70%, it is difficult to establish correlations with single biological features. It should be also emphasized that the donors have been selected upfront based on the presence of positive immunogenetic variables, thus already influencing the donor cohort. Therefore, this selection bias may be at least in part responsible of the lack of statistical correlation with clinical outcome. Here, we also evaluated if naïve versus differentiated phenotype, considering the NK cell repertoire in both donors and recipients undergoing αβT/B-cell depleted haplo-HSCT, influences the clinical outcome. Since these characteristics were not included in donor selection criteria, cluster distribution in donors did not deviate from a random choice, as it appears similar to that shown in the pivotal study, describing an unrelated donor cohort [41]. Moreover, this clustering analysis could be considered correct, since a more differentiated phenotype in the transplanted patients was associated with HCMV reactivation, consistent with a previous report in the same transplantation platform [32]. Bjorklund et al. reported that donor NK cell repertoires dominated by naïve NK cells were protective against leukemia relapse in AML and myelodysplastic syndrome (MDS) patients after RIC and HLA-matched graft, focusing on a poor anti-leukemia ability of terminally differentiated NK cells. In our cohort, we could not find any significant correlation between differentiated/naïve NK-cell phenotype, considering both donors and transplanted patients, and relapse-free survival. Stratifying patients by type of leukemia, a trend toward a better clinical outcome could be observed in AML patients receiving cluster naïve graft, while in ALL receiving cluster differentiated graft. Larger cohorts will be necessary to eventually confirm the observed trends, possibly reaching a statistical significance. In another study, Cichocki et al. reported, in patients receiving RIC and UCB, a direct association of the expansion of adaptive NK cells after HCMV reactivation associated with better LFS [53,54]. However, even if we confirmed an impact of HCMV on NK-cell maturation, in our study, we could not identify any clinical correlation between this more differentiated NK cell phenotype in transplanted patients and LFS, also differentiating by the type of leukemia. The different results in our cohort might be related to the use of myeloablative conditioning, the choice of HLA-haploidentical donors, and the presence of mature NK cells in the graft. Since adaptive NKG2C^+^ NK cells usually express a specific iKIR (particularly self-KIR2DL) and lack NKG2A, it is crucial to understand whether HCMV reactivation can trigger or not the expansion of the alloreactive subset. We found a certain degree of heterogeneity regarding the size of alloreactive NK cell subsets in the reconstituted NK cell repertoire, possibly depending on the type of alloreactivity (Table 1 and Figure 3A). During NK-cell development, the first expressed iKIR is KIR2DL2/L3 [55,56], which can frequently associate with NKG2C when early HCMV reactivation occurs. In Allo C1 transplants, sp2DL2/L3 donor-derived NK cells can be sustained. Thus, high percentages of sp2DL2/L3 NK cells were detected especially in patients who experienced HCMV reactivation and showed an adaptive phenotype. In Allo C2 transplants, the size of sp2DL1 NK cells detected in patients was almost always lower than in donors, and HCMV reactivation seemed not to be helpful in its increment or even maintenance. In Allo Bw4 transplant, the size of sp3DL1 is quite low in most donors (lower than 5%). It is possible that an HCMV-related expansion of NK cells expressing NKG2C in association with self-KIR2DL can contrast this type of NK alloreactivity. To address this issue in more detail, it will be necessary to analyze a larger cohort of αβT/B-cell depleted haplo-HSCT and a more precise identification of adaptive features, defined by KIR^+^NKG2C^+^CD57^+^ phenotype and/or lacking FcεRγ, EAT-2, and SYK expression.

The adoptive transfer of mature NK and γδT cells with the graft and their persistence in the recipient as functional effector cells not impaired by any pharmacological post-HSCT GvHD prophylaxis is crucial and can explain the clinical success of αβT/B-depleted haplo-HSCT. So far, to further improve the outcome of αβT/B-depleted haplo-HSCT, two new approaches involving either innate and adaptive immune system have been explored. The use of zoledronic acid for promoting Vδ2 cell differentiation and cytotoxicity against leukemia blasts has been recently tested in a cohort of pediatric patients [57]. A clinical trial, based on post αβT/B-depleted haplo-HSCT adoptive transfer of donor-derived T cells genetically modified with a suicide gene [58], has been conducted with the aim of accelerating recovery of adaptive immunity (NCT02065869). However, no significant influence on NK cell reconstitution has been observed [59] in this latter group. A further attempt to optimize graft engineering in the haploidentical setting might include the depletion of CD66b^+^ polymorphonuclear myeloid derived suppressor cells (PMN-MDSC), which were recently demonstrated to exert an inhibitory effect on NK cells [60]. Finally, the donor selection criteria can be continuously refined, following the increased knowledge on NK-cell mediated GvL effect, as well as on specific features correlating with better clinical outcome in transplanted patients. In fact, more recently, in the evaluation of the *KIR* genotype, we started to consider the proposed centromeric and telomeric KIR score (ct-KIR score) reported to correlate Cen B/x and Tel A/A donor *KIR* repertoires with low relapse risk in ALL pediatric patients undergoing allo-HSCT [61].

## 4. Materials and Methods

### 4.1. Patients and Donors

Our cohort refers to 80 pediatric patients with ALL (*n* = 56) and AML (*n* = 24) (Appendix A), enrolled in the phase I/II study of haplo-HSCT after TCRαβ/CD19 negative selection, from September 2011 to September 2014 with the follow-up until December 2019 [34]. This clinical trial was approved by the Ethical Committee of the Ospedale Pediatrico Bambino Gesù (OPBG, Rome, Italy, TCR αβ haplo-HSCT-OPBG; Prot. n. 424/2011) and registered at ClinicalTrial.gov website (NCT01810120). Written informed consent was obtained from patient’s parents in accordance with the Helsinki Declaration. Peripheral blood mononuclear cells (PBMC) from donors and patients at different time points after transplantation, isolated by density-gradient centrifugation, were freshly analyzed and cryopreserved in FBS containing 10% DMSO. Primary leukemia blasts were derived from peripheral blood (PB) or bone marrow of some pediatric patients at diagnosis.

### 4.2. KIR, KIR-Ligand, and NKG2C Analyses

DNA was extracted using the QIAamp DNA Blood Mini kit (QIAGEN, Hilden, Germany). Presence of KIR-L mismatch in GvH direction was evaluated analyzing high-resolution *HLA class I* typing with KIR-ligand calculator program (http://www.ebi.ac.uk/ipd/kir/ligand.html). The results were integrated considering the C1 epitope of HLA-B*46:01 and -B*73:01 [62] and the Bw4 epitope present on HLA-A*23, -A*24 and -A*32 [63]. Moreover, among the Bw4^pos^ allotypes, HLA-B*13:01, B*13:02 and HLA-A*25 were ignored because they have been demonstrated not to be KIR3DL1 ligands [63]. *KIR* gene profiles were analyzed by sequence-specific-primer-PCR (SSP-PCR) using KIR genotyping kit (GenoVision, Saltsjoebaden, Sweeden) [64]. *NKG2C* genotypes were analyzed as previously described [51].

### 4.3. NK Cell Phenotype of Donors and Post-Transplant Patients

Surface phenotype of NK cells from donors and patients after HSCT was analyzed on freshly derived or thawed PBMC by immunofluorescence. Appropriate combinations of specific antibodies were used to identify iKIR and aKIR, and various NK-cell subsets including the alloreactive one, by multi-parametric flow cytometry performed on Gallios flow-cytometer (Beckman Coulter, Brea, CA, USA) or MACSQuant-analyser (Miltenyi-Biotec, Bergisch Gladbach, Germany). All antibodies used are listed in Appendix A. Indirect labeling was performed for KIR2DL3 staining using ECM41 (anti-KIR2DL3) mAb followed by FITC-conjugated goat anti-mouse IgM (Southern Biotech, Birmingham, AL, USA). The alloreactive NK cell subsets can be identified using appropriate mAb combinations as described in Appendix A. Data were analyzed using FlowJo Version 10 (BD Biosciences, San Jose, CA, USA).

### 4.4. Leukemia Cells

The NALM-16 cell line (pediatric B cell precursor leukemia, DSMZ, Braunschweig, Germany), characterized by hemizygous HLA class-I haplotype, carrying C1 and Bw4 epitopes [65], was cultured in complete medium (RPMI 1640 supplemented with 10% heat-inactivated FBS, 2 mM L-glutamine, 100 U/mL penicillin-streptomycin).

Primary AML, BCP-ALL, and T-ALL blasts collected from PB or BM of pediatric patients at diagnosis after density-gradient centrifugation were analyzed by immunofluorescence and cryopreserved. Five samples that were selected on the basis of KIR-L expression, displayed ≥90% of leukemia blasts, as documented by cytofluorimetric analysis. In addition, leukemias were also characterized for the expression levels of activating receptors ligands (i.e., DNAM-1-L, NKG2D-L, and NKp30-L) and of HLA class-I molecules (Appendix A and Appendix A) by indirect immunofluorescence and cytofluorimetric analysis using specific mAb and appropriate secondary reagents (Southern Biotech). Staining index, used to indicate expression levels, is calculated as the difference between the median fluorescence intensity of cells stained with the relevant mAb and that of the negative control divided by two times the standard deviation of the negative control.

### 4.5. Functional Assays

Standard 4h ^51^Cr-release assays (CRA) were performed using IL-2 activated polyclonal NK cell populations obtained from healthy donors, as effector cells, and leukemia target cells, at E:T ratio 10:1. Thawed leukemia blasts were labeled overnight with ^51^Cr (Perkin Elmer, Waltham, MA, USA). In masking experiments [21,33], saturating amounts of the following mAb, either alone or in combination, were added: KL247 (IgM, anti-NKp46), KS38 (IgM, anti-NKp44), F252 (IgM, anti-NKp30), F5 (IgM, anti-DNAM-1), and BAT221 (IgG1, anti-NKG2D).

In degranulation assays, 2 × 10^5^ PBMC obtained from HSCT donors or transplanted patients, cultured 3–5 days in the presence of IL-2 (600IU/mL), were incubated for 3 h either in the absence or in the presence of 1 × 10^5^ target cells with the addition of PE-conjugated anti-CD107a mAb. Thawed leukemia blasts, used as targets, were rested overnight in complete medium at 4 °C before the co-culture with effector cells. Golgi Stop (BD Biosciences Pharmingen) was added after the first hour of incubation. Thereafter, cells were collected, washed in PBS with 2% FBS and 2mM EDTA and stained with anti-CD3, -CD56, -CD107a mAbs, and the appropriate antibody combinations allowing the identification of NK cell subsets (Appendix A). Samples were analyzed by Gallios flow-cytometer (Beckman Coulter Brea, CA, USA). Data analysis was performed using FlowJo Version 10 (BD Biosciences, San Jose, CA, USA); gating strategy is shown in Appendix A. Data referred as ΔCD107 represent the difference between the % of CD107a^+^ NK cells following target stimulation and the % of CD107a^+^ NK cells after incubation with medium alone.

### 4.6. Unsupervised Hierarchical Cluster Analysis

The NK immunophenotyping data from flow cytometry including the percentages of CD56^bright^, NKG2A^+^, CD57^+^, NKG2C^+^, and KIR^+^NKG2A^−^ were collected for both donors and recipients at 3–6 months after transplant. The unsupervised hierarchical clustering was performed as previously described by Bjorklund et al. [41] adopting the R package pheatmap (version 1.0.8) setting “Ward” as clustering method and “Manhattan” as distance metric. The data were normalized computing a Z-score normalization and scaled on samples (rows). The results were showed as heatmap with the corresponding annotations on the top.

### 4.7. Statistical Analysis

Three or more groups of paired or not paired samples were analyzed with Friedman or Kruskal–Wallis tests, respectively. Dunn’s post-test was used for multiple comparison. Two groups of paired samples were compared with the two-tailed Wilcoxon Mann–Whitney non-parametric test. Graphic representations and statistical analysis were performed using GraphPad Prism 6 (GraphPad Software, Inc. La Jolla, CA, USA). In addition, *p*-values above 0.05 were considered not significant.

Survival analysis (OS and LFS) was performed using the Kaplan–Meier method (R package *survival*, functions *survfit* and *survdiff*). The cumulative incidence of relapse was analyzed adopting the Fine and Gray’s (FG) implemented in the *cmprsk* R package.

## 5. Conclusions

The present study provides the first in-depth evaluation on the NK-cell receptor repertoire (combining genotypic and phenotypic profiles) in donor/recipient pairs of αβT/B-cell depleted haplo-HSCT whose clinical data have been already reported (clinical trial NCT01810120) [34]. The donor selection process was performed by applying an algorithm based on NK-cell characteristics known to be associated with a greater anti-leukemia activity. While designing this algorithm, we hypothesized that the selection of the more appropriate donor is particularly relevant in the αβT/B-cell depleted haplo-HSCT platform, since NK cells are adoptively infused with the graft. In vitro functional assays against pediatric primary leukemia cells clearly documented that the alloreactive NK cell subset is endowed with the highest anti-leukemia activity, supporting the benefit of selecting haploidentical donors with NK alloreactivity. The evidence of the primary role of NCR in leukemia cell recognition and lysis induction was also shown, with NKG2D and DNAM-1 being minor contributors. In addition, we provided a detailed analysis of NK-cell repertoire on the different stages of reconstitution. We demonstrated that, already one month after HSCT, donor-derived mature and functional NK cells circulate in the recipients’ peripheral blood, including the alloreactive and/or adaptive subsets. Conversely, at this early time point, NK cells derived from HSCs display an immature phenotype. At a later stage, through unsupervised hierarchical clustering, we identified a naïve and differentiated pattern of NK-cell phenotype. The occurrence of HCMV reactivation was associated with a more differentiated NK-cell phenotype, variably affecting the size of alloreactive subsets. However, the two different clusters did not identify subgroups with differences in OS, LFS, or relapse incidence, but the investigation in a bigger cohort of αβT/B-depleted haplo-HSCT recipients will be necessary to better elucidate this issue.

## Figures and Tables

**Figure 1 cancers-12-02187-f001:**
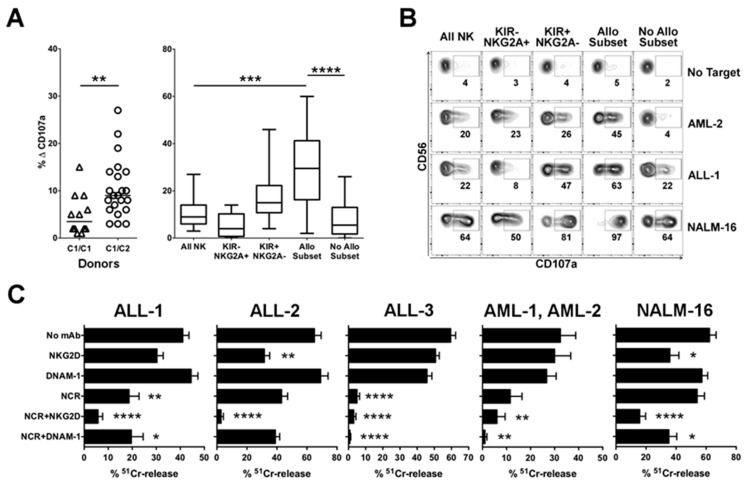
Relevance of alloreactivity and NCR for NK-mediated anti-leukemia activity. (**A**) NK cell specific degranulation activity (ΔCD107a) after stimulation with HLA-C1/C1 primary ALLs (ALL-1 and ALL-3). E:T ratio 2:1. Left: comparison of HLA-C1/C1 (C1/C1) (*n* = 12) and HLA-C1/C2 (C1/C2) (*n* = 22) donors. ** *p* < 0.01 (Mann–Whitney test). Horizontal bars indicate the medians. Right: different NK cell subsets derived from the HLA-C1/C2 donors are compared. The Allo Subset consists on sp2DL1 NK cells, No Allo Subset represents NK cells expressing iKIRs specific for ligands on target cells. *** *p* < 0.001, **** *p* < 0.0001 (Kruskal–Wallis test with Dunn’s post-test). Whisker lines represent the highest and lowest values; horizontal lines represent the medians. (**B**) Degranulation capacity of different NK cell subsets derived from a representative donor (donor of UPN58) in the absence or in the presence of the indicated targets. E:T ratio 2:1. Numbers indicate the percentage of surface CD107a^+^ cells. Allo Subset consists on sp2DL1 NK cells, No Allo Subset represents NK cells expressing iKIRs specific for ligands on target cells. (**C**) Polyclonal NK cell populations from healthy donors were tested to evaluate the involvement of different activating receptors in the lysis of ^51^Cr-labeled ALL-1 (DNAM-1-L^−^, NKG2D-L^−^), ALL-2 (DNAM-1-L^−^, ULBP-1^+^), ALL-3 (Nectin-2^+^, NKG2D-L^−^), AML-1 (PVR^+^, Nectin-2^+^, NKG2D-L^−^), AML-2 (Nectin-2^+^, NKG2D-L^−^) and NALM-16 cell line (PVR^+^, Nectin-2^+^, MICA^+^, ULBP-1^+^, ULBP-2^+^, ULBP-3^+^) (see also Appendix A and Appendix A). E:T ratio 10:1. NK cells were pre-incubated with saturating amounts of mAb to the indicated receptors. Data are shown as percentage of lysis, mean + SEM. Each target has been tested with effector cells derived from at least *n* = 7 donors. Three independent experiments were performed. Significant differences between lysis in the presence of blocking mAb in comparison to “No mAb” are shown: * *p* < 0.05; ** *p* < 0.01; **** *p* < 0.0001 (Friedman test with Dunn’s post-test).

**Figure 2 cancers-12-02187-f002:**
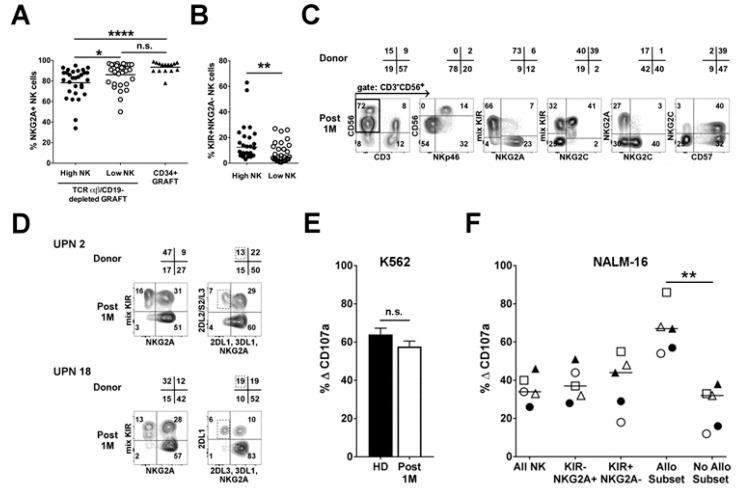
Evidence of mature and functional donor-derived NK cells in the patient repertoire at one month after HSCT. Cytofluorimetric analyses of NK cells (CD3^−^CD56^+^ gated peripheral blood mononuclear cells, PBMC) derived from patients at 1 month after αβT/B-depleted haplo-HSCT. (**A**) Percentages of NKG2A^+^ cells in patients who received either high (*n* = 30) or low (*n* = 30) NK cell numbers (> or <34.6 × 10^6^/kg, respectively), in comparison with CD34^+^ (*n* = 15) engrafted patients. * *p* < 0.05, **** *p* < 0.0001 (Kruskal–Wallis test with Dunn’s post-test). Horizontal bars indicate the medians. (**B**) Percentages of KIR^+^NKG2A^−^ cells in patients who received grafts containing either high (*n* = 26) or low (*n* = 30) NK cell numbers. ** *p* < 0.01 (Mann–Whitney test). Horizontal bars indicate the medians; (**C**) analysis of NK-cell repertoire in a representative patient (UPN77) infused with high cell numbers of NK cells characterized by an “adaptive” phenotype. The respective percentages of donor NK cells are also reported over the plots; (**D**) NK cell expression of KIR and NKG2A (left) and detection of alloreactive subset (right) in two representative patients, both infused with high NK cell numbers, are shown. The corresponding percentages of donor NK cells are also reported over the plots. The alloreactive subset is indicated as a dotted square. (**E**) Degranulation activity against K562 cells of NK cells from patients at one month after haplo-HSCT (post 1M, *n* = 6) in comparison to NK cells from healthy donors (HD, *n* = 27). E:T ratio 2:1. Data are shown as mean + SEM; n.s., no significant *p*-values (Mann–Whitney test). (**F**) Degranulation activity against NALM-16 of NK cells from patients (each identified by a different symbol) at 1 month after haplo-HSCT. E:T ratio 2:1. Different NK cell subsets, including the Allo C2 NK cell subset (Allo Subset), are compared. No Allo Subset represents NK cells expressing iKIRs specific for ligands on target cells. Horizontal bars indicate the medians. ** *p* < 0.01 (Kruskal–Wallis test with Dunn’s post-test).

**Figure 3 cancers-12-02187-f003:**
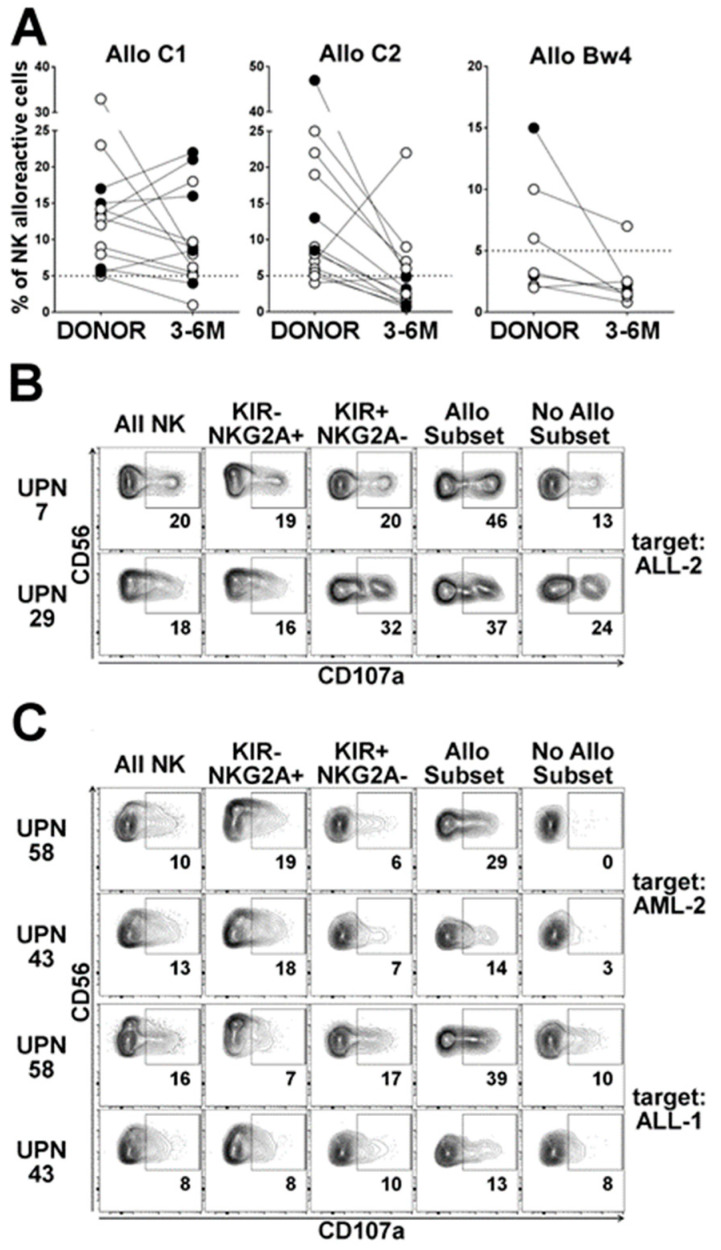
Presence of donor-derived alloreactive NK cell populations in patients after haplo-HSCT. (**A**) Percentages of alloreactive NK cell subsets (gating on CD3^−^CD56^dim^) in patients 3–6 months (3–6 M) after haplo-HSCT were compared with those observed in the donors, considering the different type of alloreactivity. Filled black circles refer to donor/recipient pairs in which patients experienced HCMV reactivation. The dotted lines are set at 5%, considered a cut-off for a good size of alloreactive subset; (**B**,**C**) degranulation activity of NK cells from patients at six months after haplo-HSCT against pediatric leukemia blasts. E:T ratio 2:1. CD107a expression was evaluated on CD3^−^CD56^+^ NK cells (All NK) and on different NK cell subsets, as indicated. Allo Subset represents the alloreactive NK cell subset, No Allo Subset consists of NK cells expressing iKIRs specific for ligands on target cells. Numbers represent the ΔCD107a. In (**B**), NK cells from two representative patients, transplanted with Allo C1 donors, were co-cultured with HLA-C2/C2 ALL-2. In (**C**), NK cells from two representative patients, transplanted with Allo C2 donors, were co-cultured with HLA-C1/C1 AML-2 or ALL-1.

**Figure 4 cancers-12-02187-f004:**
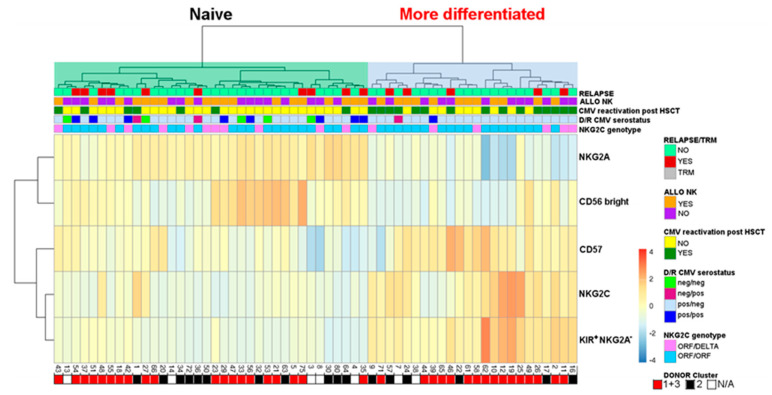
Hierarchical clustering of post-transplant repertoires. Unsupervised hierarchical clustering of NK-cell repertoires in 60 patients at 3–6 months after HSCT allowed the identification of two main clusters, namely cluster 1 with “naïve” and cluster 2 with a more differentiated repertoire. Data were based on multi-color flow-cytometry, analyzing the frequencies of CD56^bright^, or, among the CD56^dim^, the frequencies of NKG2A^+^, CD57^+^, NKG2C^+^, and KIR^+^NKG2A^−^ NK cells. Information on clinical and biological observations is annotated in the top bars above and to the right of the heat map. Z-scores were calculated and used to scale for visualization across rows in the heat map. High (red) and low (blue) frequencies of each subset are represented in the color scale. Underneath the heatmap, red and black squares indicate the cases whose respective donors showed NK cells with “differentiated” (identified by cluster 1 + 3) and “naïve” (identified by cluster 2) phenotype, respectively, as shown in Appendix A, while white squares indicate when there is no available donor data (NA).

**Figure 5 cancers-12-02187-f005:**
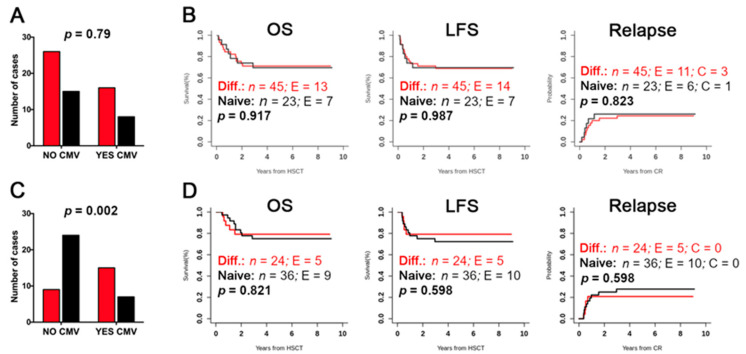
Correlation analysis of clusters with clinical outcome. The clusters identified in donors (*n* = 68) (**A**,**B**) and patients at 3–6 months after HSCT (*n* = 60) (**C**,**D**) have been analyzed for possible association with HCMV reactivation, OS, LFS, and cumulative incidence for Relapse. While the differentiated (red columns) versus naïve (black columns) phenotype of donor NK cells (**A**) did not predispose to HCMV reactivation after haplo-HSCT (*p* = 0.79), the differentiated phenotype of post-transplant NK cells (**C**) was highly associated with HCMV reactivation (** *p* = 0.002). Significance was calculated by the χ^2^ test. OS, LFS, and Relapse according to differentiated and naïve NK-cell repertoires identified in donors (**B**) and post-HSCT patients (**D**) are shown. Fischer’s exact text. No significant *p*-values were detected.

**Table 1 cancers-12-02187-t001:** Description of cases transplanted from NK alloreactive donors.

ALLO *	UPN	Diagnosis	Donor	HCMV Serology D/R	*NKG2C*	B cont	Permissive iKIR *	aKIR (2DS1, 2DS2) ^#^	Viral Infections ^∞^	Acute GvHD	Relapse/TRM	Alive	Infused NK Cells (×10^6^/kg) ^§^	Alloreactive Subset ^‡^			Cluster Phenotype ^¶^	
														Donor	Recipient Post-1M	Recipient Post-3/6M	Donor	Recipient
C1	1	AML (M4)	Father	neg/pos	ORF/ORF	2	2DL2/L3	**2DS1**,2DS2	**HCMV**	NO	NO	YES	33.9	5	2	8	Naive	Naive
	2	T-ALL	Mother	pos/pos	ORF/ORF	1	2DL2/L3	2DS2	**HCMV**	NO	NO	YES	**82.83**	13	7	21	Naive	Adaptive
	6	BCP-ALL	Father	neg/pos	ORF/ORF	1	2DL3	**2DS1**	NO	NO	Relapse	NO	**75.3**	12	5	18	nd	nd
	7	BCP-ALL	Mother	neg/pos	ORF/ORF	1	2DL3	**2DS1**	**HCMV**	NO	NO	YES	**65.95**	17	nd	22	nd	Adaptive
	10	BCP-ALL	Mother	pos/pos	ORF/ORF	1	2DL2/L3	2DS2	BK	NO	NO	YES	**94.76**	9	nd	5	Adaptive	Adaptive
	22	BCP-ALL	Father	pos/pos	ORF/ORF	1	2DL2/L3	2DS2	NO	Grade II	NO	YES	22.4	8	4	5	Naive	Adaptive
	29	AML (M1)	Father	pos/neg	ORF/delta	3	2DL2/L3	**2DS1**,2DS2	NO	NO	NO	YES	22.6	14	2	9	Naive	Naive
	35	T-ALL	Mother	pos/neg	ORF/ORF	2	2DL2	2DS2	NO	Grade I	Relapse	NO	16.4	23	2	6	Naive	Naive
	40	T-ALL	Father	pos/pos	delta/delta	0	2DL3	NO	NO	NO	NO	YES	13.4	33	10	8	NA	NA
	45	AML (M7)	Mother	pos/pos	ORF/ORF	1	2DL3	**2DS1**	**HCMV**, ADV	NO	Relapse	NO	**56.1**	20	25	NA	Naive	NA
	38	AML (M4)	Mother	pos/pos	ORF/ORF	0	2DL3	NO	**HCMV**, VRS	NO	NO	YES	23.3	15	15	16	nd	Adaptive
	73	B-ALL	Father	pos/pos	ORF/delta	3	2DL2/L3	**2DS1**,2DS2	HHV6	Grade II	TRM	NO	30	20	nd	NA	Naive	NA
	75	AML (M7)	Father	pos/pos	ORF/ORF	1	2DL2/L3	2DS2	NO	NO	Relapse	NO	**57.5**	5	0	1	Adaptive	Naive
	5	T-ALL	Father	pos/pos	ORF/ORF	2	2DL2/L3, 3DL1	**2DS1**	BK	NO	NO	YES	**86.6**	13	4	9	Naive	Naive
	64	B-ALL	Mother	pos/pos	ORF/delta	2	2DL3, 3DL1	**2DS1**,2DS2	**HCMV**, BK	NO	Relapse	NO	**57.6**	6	4	4	Naive	Naive
C2	4	BCP-ALL	Mother	pos/neg	ORF/ORF	0	2DL1	NO	NO	NO	NO	YES	**53.9**	5	1	1	nd	Naive
	18	BCP-ALL	Father	pos/pos	ORF/ORF	1	2DL1	2DS1	NO	NO	NO	YES	**87.9**	19	6	7	Naive	Naive
	23	BCP-ALL	Mother	pos/pos	ORF/delta	0	2DL1	NO	**HCMV**	Grade II	NO	YES	16.7	13	5	3	Naive	Naive
	24	AML (M2)	Father	pos/pos	ORF/ORF	2	2DL1	2DS1,**2DS2**	BK	NO	Relapse	NO	**66.7**	(7)	(15)	(22)	nd	nd
	27	AML (M0)	Mother	neg/neg	ORF/ORF	1	2DL1	**2DS2**	NO	NO	Relapse	NO	21.5	(9)	(1.6)	(2)	Naive	Naive
	30	BCP-ALL	Mother	pos/pos	ORF/ORF	0	2DL1	NO	NO	Grade I	NO	YES	13.6	6	0.7	1	Naive	Naive
	39	BCP-ALL	Father	pos/neg	ORF/delta	3	2DL1	2DS1,**2DS2**	NO	NO	NO	YES	**60.2**	(4)	(2.5)	(5)	Naive	Adaptive
	43	AML (M4)	Father	pos/pos	ORF/delta	0	2DL1	NO	**HCMV**	NO	NO	YES	**54.1**	47	4	5	Adaptive	Naive
	50	AML (M4)	Father	pos/pos	ORF/delta	3	2DL1	2DS1,**2DS2**	NO	NO	NO	YES	6.8	(5)	(0)	(1)	Naive	Naive
	58	AML (M5)	Father	pos/pos	ORF/delta	1	2DL1	**2DS2**	NO	Grade II	NO	YES	31.2	(22)	(5)	(6)	Adaptive	Adaptive
	61	B-ALL	Mother	pos/pos	ORF/ORF	1	2DL1	**2DS2**	H1N1	NO	NO	YES	**51.67**	(25)	(7)	(9)	Adaptive	Adaptive
	66	T-ALL	Mother	pos/pos	ORF/ORF	3	2DL1	2DS1,**2DS2**	HHV6	Grade I	NO	YES	12.77	(8)	(3)	(2)	Naive	Naive
	71	BCP-ALL	Mother	pos/pos	ORF/ORF	1	2DL1	**2DS2**	**HCMV**, HHV6	NO	NO	YES	12.4	(8)	nd	(1)	Adaptive	Adaptive
	79	AML (M4)	Mother	pos/neg	ORF/ORF	0	2DL1	NO	NO	NO	NO	YES	28.3	6	1	nd	Naive	nd
Bw4	12	BCP-ALL	Mother	pos/pos	ORF/ORF	1	3DL1	**2DS2**	**HCMV**	Grade II	NO	YES	**90.2**	(15)	(2)	(2)	Naive	Adaptive
	26	BCP-ALL	Father	pos/pos	ORF/ORF	2	3DL1	**2DS2**	**HCMV**	NO	Relapse	NO	35	(3)	(2)	(2)	Adaptive	Adaptive
	20	AML (M3)	Mother	pos/pos	ORF/delta	3	3DL1	**2DS1,2DS2**	NO	NO	NO	YES	**82.3**	(6)	nd	(2)	Naive	Naive
	21	BCP-ALL	Mother	pos/pos	ORF/ORF	2	3DL1	**2DS1,2DS2**	NO	NO	NO	YES	**36.5**	(3)	nd	(2)	Naive	Naive
	47	ALL-bifen	Mother	pos/pos	ORF/ORF	0	3DL1	NO	NO	NO	NO	YES	7.87	2	1	1	Naive	Naive
	51	T-ALL	Mother	pos/neg	ORF/ORF	0	3DL1	NO	NO	NO	NO	YES	**44.7**	10	2	7	Adaptive	Naive
	72	T-ALL	Father	pos/pos	ORF/delta	2	3DL1	2DS1,**2DS2**	ADV	Grade II	NO	YES	**82.6**	(2)	nd	(2)	Naive	Naive

UPN, unique patient number; BCP-ALL, B-cell precursor ALL; D/R, donor/recipient; HCMV, human cytomegalovirus; ADV, adenovirus; BK, BK virus; VZV, varicella zoster virus; HHV6, human herpesvirus 6; H1N1, H1N1 influenza virus; TRM, transplant related mortality; B cont, B content; nd, not determined; NA, not applicable, in case of impossibility of patient’s sampling (because of TRM or early relapse), or lack of NKG2C (*NKG2C^del/del^* donor), representing a relevant marker for cluster phenotype. * All cases are grouped in Allo C1, Allo C2, and Allo Bw4, considering the KIR-L present in the donor and missing in the recipient and the presence of donor iKIR specific for the mismatched KIR-L (identified as “Permissive iKIR” in the appropriate column). In Allo C1 group, UPN5 and UPN64 showed also Bw4 mismatch. ^#^ Among the aKIR, only the presence of KIR2DS1 and KIR2DS2 in donor’s *KIR* genotype has been reported. 2DS1 is in bold when HLA-C1^+^ donor and HLA-C2^+^ patient (i.e., E/U), 2DS2 is in bold when it can contribute to the alloreactive subset, but cannot be quoted by flow-cytometry. ^‡^ The percentage of alloreactive subset was evaluated in peripheral blood NK cells (gating on CD3^−^CD56^+^ cells) of donors and post-HSCT patients at 1 month (Recipient post-1M) or 3/6 months (Recipient post-3/6M). Numbers are in brackets when the size of the alloreactive subset might be underestimated by the presence of KIR2DS2. **^∞^** Occurrence of viral infection after HSCT; HCMV reactivation is in bold. **^§^** Numbers in bold when higher than the median value (i.e., 34.6 × 10^6^/kg body weight). ^¶^ Cluster phenotype has been determined for donors and for post-HSCT patients (see Section 2.4).

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
