# Peer review of "Phenotypic and Functional Characterization of NK Cells in αβT-Cell and B-Cell Depleted Haplo-HSCT to Cure Pediatric Patients with Acute Leukemia"

_cancers, 2020, doi:10.3390/cancers12082187_

Round 1
Reviewer 1 Report
Meazza and colleagues characterized the NK-cell receptor repertoire in pediatric acute leukemia patients treated with αβT/B-depleted haplo-HSCT. This phenotypical and functional analysis was based on the results of the clinical trial #NCT01810120 which reported excellent clinical outcome after this treatment modality.
They demonstrate that functional alloreactive NK cells are circulating in the peripheral blood of the recipients already after 1 month with an increase up to 3 to 6 months. They further report that these alloreactive NK cells show potent anti-leukemia activity. They successfully show that donor/recipient receptor-ligand mismatch for inhibitory KIRs is correlated with a better clinical outcome thereby providing confirmative evidence for previous results and demonstrate that this results in higher proportions of mature and functional NK cells.
The manuscript is well written and the experiments are designed and conducted appropriately. The results are presented comprehensible and the conclusions are drawn carefully and convincingly. The manuscript expands the knowledge regarding NK cell biology in the context of HSCT, highlighting the role of inhibitory KIRs and KIR mismatch in this context.
The only issue that I would like to raise is that the study would largely benefit from inclusion of the data showing the expression of the ligands for the functionally investigated activating NK receptors on ALL target cells
Reviewer 2 Report
In this manuscript, Raffaella Meazza et al. analyzed in detail the NK cells in pediatric patients with acute leukemia underwent hematopoietic stem cell transplantation in αβ T-cell and B-cell depleted haploidentical protocol. The authors justify the multistep algorithm of donor selection based on allreactivity mediated by the mismatch of donor self inhibitory KIR and recipient HLA-I as the most important criterion and provide the impressively detailed analysis of NK cell repertoire on the different stages of the reconstitution and clinical results. This essential information is in great demand by potential readers. The results of the work are addressed the burning questions of impact of NK cell repertoire, both donor and recipient, on HCMV reactivation and the clinical outcome. Even the authors did not find any association between the more differentiated NK cell phenotype and overall survival or incidence of relapse, this work is of great relevance of publishing in Cancers.
Next concerns may be addressed to improve the overall quality of the manuscript.
- This is an excellent work but written in a very complicated manner. Despite the complexity of the subject, the authors are encouraged to present the material not in a so heavy form, especially, in section 2.1 and 2.3, concerning donor selection criteria and analysis of reconstituted repertoire of NK cells after transplantation. The style of the result section may be considerably improved.
- Since in the conditions of the haplo-HSCT the αβ T cells were depleted, it would be of interest to provide data on and discuss more intensively the gd T cell compartment size in both donor and recipient reconstituted lymphocyte repertoire.
Reviewer 3 Report
The authors present an in-depth evaluation on NK cell subpopulations and function in pediatric leukemia patients treated with haploidentical HSCT. In this setting alpha/beta T cells and B cells are depleted with the GVL mostly being mediated by NK cells and gamma delta T cells. The patient cohort and clinical outcomes were previously reported in Blood in 2017 and this study probes in great depth into NK cell subsets in these transplant recipients and evaluates the persistence of donor NK subsets with additional functional studies. The manuscript is well written and is notable for the relatively large pediatric patient cohort analyzed. Further, the immunophenotypic assessment of NK cells and evaluation of how donor KIR ligands and HLA type of donor and recipient impact alloreactivity is expertly designed, presented and analyzed. The authors demonstrate that alloreactive defined NK cell subsets have higher degree of degranulation relative to non-alloreactive subsets. Further they demomstrate that NCRs are the primary mediators of target cells recognition, with NKG2D and DNAM-1 being minor contributors. They also demonstrate persistence of donor NK cells subsets at one month and 3-6 months with maintenance of relevant cytotoxic function.
The authors used unsupervised hierarchical clustering to identify a naive and differentiated pattern of NK cell differentiation, but these clusters do not identify subgroups with differences in OS, LFS or relapse. However, they do demonstrated that HCMV reactivation was associated with a more differentiated NK cell immunophenotype.
There are some minor editorial issues that should be address and are listed below:
1) Page 1, Abstract, line 30: use of 'major players' should be changed to a more scientific term (e.g. NK cells can exert relevant graft-verus leukemia effects in haplotype transplants for leukemia).
2) Page 2, Introduction, line15: The original works demonstrating clinically relevant improvements in survival in haplotype transplants with KIR mismatches are cited (ref 3 and 4), but no recent studies which did not demonstrate similar benefits. The authors should select a few of these negative studies to put these original studies in context and set up the rest of the paper for the reader. A short paragraph might be indicated here which better reflects the conflicting clinical studies in the field.
3) Page 3, Results, 2.1, line 14 and 15; Please provide reference for criteria 'd' higher absolute levels of NK and gamma delta T cells
4) Page 7, Resuts, Line 14: Has LFS been defined earlier in full- please check'
5) Page 7, Results, Line 31-32: "...cell line resulted to be highly susceptible" check grammar here.
6) Page 8, Results, Line 11: delete 'also' in current position and move to end of sentence to read: "was also preserved."
7) Page 9, Results, Line 1: please add E:T ratio here. It appears in the methods as 10:1. Would be helpful if this is in the legend as well.
8) Page 10, Results, Line 10: change "consists on" to "consists of".
9) Page 11, Results, Line15: change "Differently" to "In contrast".
10) Page 12, Discussion, Line 17: change "manipulation's" to "manipulation"
11) Page 12, Discussion, Line 30: has NRM been defined earlier- please check. Best to review all abbreviations in paper and check if they are written in full on first instance.
12) Page 13, Discussion, Line 17: has PT-Cy been defined previously- please check.
13) Page 14, Discussion, Line 14: has RIC been defined previously- please check
14) Page 14, Discussion, Line 49: has PMN-MDSC been defined previously- please check.
15) Page 16: Conclusion, Lines 37-45: The conclusion is not particularly well written relative to the Discussion and needs to be revised. It should recapitulate the major novel and confirmatory findings of the paper (some listed in my opening paragraph). The last sentence should be removed and the future directions element of this paragraph should be revised. Since the paper has a lot of highly technical immunology many readers will look at this section, so it should be given careful attention to emphasize the major points of this excellent work.
Reviewer 4 Report
I think this is the informative manuscript for pediatric leukemia patients for donor selection.
・The author evaluated various disorder such as BCP-ALL, T-ALL, and AML. It is meaningful , but evaluating each one would be better.
・I don't think I clearly understood the consideration about HCMV.
・As the author mentioned in the manuscript, I think PTCY would be an important GVHD prophylaxis in the future. If the author could provide the additional data about PTCY, I think it would be better.
Though I am following the opinions of other reviewers, I think it would be possible to accept this manuscript after revision.
